# Necroptosis as a Novel Facet of Mitotic Catastrophe

**DOI:** 10.3390/ijms23073733

**Published:** 2022-03-29

**Authors:** Aleksandra Yu. Egorshina, Alexey V. Zamaraev, Vitaliy O. Kaminskyy, Tatiana V. Radygina, Boris Zhivotovsky, Gelina S. Kopeina

**Affiliations:** 1Faculty of Medicine, MV Lomonosov Moscow State University, 119991 Moscow, Russia; egorshina.aleksandra.2012@post.bio.msu.ru (A.Y.E.); a-zamaraev@yandex.ru (A.V.Z.); boris.zhivotovsky@ki.se (B.Z.); 2Division of Toxicology, Institute of Environmental Medicine, Karolinska Institute, P.O. Box 210, 171 77 Stockholm, Sweden; vitaly.kaminsky@ki.se; 3Federal State Autonomous Institution “National Medical Research Center for Children’s Health” of the Ministry of Health of the Russian Federation, 119296 Moscow, Russia; tvradigina@mail.ru

**Keywords:** mitotic catastrophe, necroptosis, autophagy, doxorubicin treatment

## Abstract

Mitotic catastrophe is a defensive mechanism that promotes elimination of cells with aberrant mitosis by triggering the cell-death pathways and/or cellular senescence. Nowadays, it is known that apoptosis, autophagic cell death, and necrosis could be consequences of mitotic catastrophe. Here, we demonstrate the ability of a DNA-damaging agent, doxorubicin, at 600 nM concentration to stimulate mitotic catastrophe. We observe that the inhibition of caspase activity leads to accumulation of cells with mitotic catastrophe hallmarks in which RIP1-dependent necroptotic cell death is triggered. The suppression of autophagy by a chemical inhibitor or *ATG13* knockout upregulates RIP1 phosphorylation and promotes necroptotic cell death. Thus, in certain conditions mitotic catastrophe, in addition to apoptosis and autophagy, can precede necroptosis.

## 1. Introduction

The various types of genomic aberrations, such as gene mutations, deletions/amplification, or chromosome segment rearrangements, often cause disturbances in the cell cycle control and repair system, allowing continuous cell division and accumulation of aneuploid cells. Mammalian cells possess a variety of mechanisms to eliminate genomically unstable cells. Mitotic catastrophe is considered as one of defensive mechanisms that senses mitotic damage and prevents the survival and/or propagation of mitotic-defective cells. This status is not the form of programmed cell death (PCD), rather it is considered as a preliminary stage of the cell self-elimination via apoptosis, necrotic-like death, or autophagy-dependent cell death [1]. Furthermore, survival exit of mitotic catastrophe could be the “mitotic slippage” in which cells return to interphase without cellular division as polyploid cells or/and cellular senescence [2].

Mitotic catastrophe is morphologically characterized by the formation of giant multinucleated or micronucleated cells. The multinucleated cells arise from clusters of mis-segregated uncondensed chromosomes, whereas micronucleated cells arise from lagging chromosomes or chromosome fragments during anaphase that stayed outside of the daughter nuclei [3]. The prominent biochemical feature of mitotic catastrophe is mitotic arrest, while the mechanisms that dictate cell fate following mitotic catastrophe remain unclear. The evasion of mitotic catastrophe could lead to premature exit from cellular mitosis and reversal to interphase without undergoing proper chromosome segregation and cytokinesis, yielding polyploid multinucleated cells. This phenomenon also known as a “spindle checkpoint adaptation” or “mitotic slippage”. Such cells arrested in interphase could enter cellular senescence causing Sting-mediated inflammation or die after slippage [4,5]. Usually, mitotic catastrophe is triggered by a low dose of chemotherapeutic agents, which is insufficient to induce cell death. Moreover, cancer cells, which are resistant to apoptosis, might undergo this condition which can finalize the cell fate as death [6]. Thus, the induction of mitotic catastrophe appears to be a perspective strategy to sensitize tumor cells for chemotherapy.

To stimulate mitotic catastrophe, a variety of agents that disrupt mitotic progression or induce DNA damage are currently used [7,8,9]. Several studies demonstrate that the anthracycline antibiotic, doxorubicin, which acts as a stabilizer of Topoisomerase II-DNA complex preventing religation of DNA double-strand breaks during replication, could stimulate mitotic catastrophe at low doses [10,11]. In doxorubicin-treated cells, mitotic catastrophe could prime cells for apoptosis, triggering p53 expression, cytochrome *c* release, caspase activation, and substrate cleavage. At the same time, in colorectal carcinoma cell lines, autophagy modulation has been demonstrated to be an essential step for cell death induction after mitotic catastrophe [11]. The authors proposed that the level of Bcl-2 (B-cell lymphoma 2) family proteins plays a significant role in the duration of mitotic catastrophe and in a crosstalk between autophagy and apoptosis. Thus, an inhibition of apoptosis by overexpression of anti-apoptotic proteins Mcl-1 or Bcl-XL promoted autophagy upon mitotic catastrophe development [6]. However, the exact mechanism of this interplay, as well as the role of other types of PCD in the completion of mitotic catastrophe, are largely unknown. For example, necrotic-like cell death has been identified as a consequence of mitotic catastrophe, but the type of this death is still unclear. Many types of PCD result in necrotic morphology: necroptosis, MPTP (Mitochondrial Permeability Transition Pore)-driven necrosis, parthanatos, ferroptosis, or pyroptosis [12]. Consequently, mitotic catastrophe can potentially lead to any of them.

In our study, we identified that the induction of mitotic catastrophe upon 600 nM doxorubicin treatment in three different cell lines led to accumulation of cells with mitotic catastrophe hallmarks in which autophagy or apoptotic and RIP1-dependent necroptotic cell death were triggered. RIP1-dependent necroptosis is well-known caspase-independent necrosis-like cell death, which could be initiated upon DNA damage through the formation of Ripoptosome complex. Ripoptosome platform consists of caspase-8/RIP1(receptor interacting protein 1)/FADD (Fas-associated protein with death domain)/cFLIP (FADD-like IL-1β-converting enzyme) [13,14]. Upon stress conditions Ripoptosome triggers phosphorylation of RIP1, RIP3 and MLKL (Mixed lineage kinase domain-like pseudokinase) to transmit the cell death signal [15,16,17]. Phosphorylation of MLKL leads to protein oligomerization and translocation to the plasma membrane where it participates in pore formation that leads to loss of plasma membrane integrity, translucent cytosol, and swelling mitochondria [15,18,19]. In current study, we demonstrated that an accumulation of necroptotic effector phospho-MLKL was detected in cells possessing mitotic catastrophe morphology that indicated triggering necroptosis after mitotic catastrophe. Additionally, the repressing of autophagy by chemical inhibitor bafilomycin A1 or knockout of *ATG13* (Autophagy-related protein 13) resulted in upregulation of RIP1 phosphorylation and necroptosis induction. 

## 2. Results

### 2.1. Appearance of Mitotic Catastrophe upon Treatment with a Sublethal Dose of Doxorubicin

To investigate the process of mitotic catastrophe mutant p53 ovarian carcinoma (Caov4), non-small cell lung carcinoma (U1810) cell lines and wild-type p53 colorectal carcinoma (HCT116) cell line were treated with 600 nM of a DNA-damaging drug, doxorubicin, for 48 h according to previously published data [11,20]. The doxorubicin treatment resulted in significant changes in nuclear morphology of all studied cell lines, characterized by an increase in the number of cells with multi- and micronucleation, the morphological hallmark of mitotic catastrophe (Figure 1A). The nuclei specific for the mitotic catastrophe were detected in 23%, 18%, and 20%, of Caov4, HCT116, and U1810 cells, respectively (Figure 1B).

### 2.2. Necroptosis as a Consequence of Mitotic Catastrophe

Mitotic catastrophe is considered as a pre-stage that senses mitotic failure and leads to cell death [20]. To investigate the type of cell death after mitotic catastrophe, the level of cell death markers was analyzed by Western Blot. In response to doxorubicin treatment, the activated fragments of caspase-3 as well as a cleavage form of PARP-1 (Poly (ADP-ribose) polymerase 1), which is a downstream target of caspase-3, were detected in all analyzed cell lines, suggesting the activation of caspase cascade and apoptosis execution in these cells (Figure 2A). Notably, in the non-tumor human kidney cell line HEK293T the same concentration of doxorubicin did not induce essential caspase-3 activation and cleavage of PARP that was observed in tumor cell lines (Appendix A). In Caov4 cells, autophagy induction was also observed, characterized by the accumulation of the lipidated form of LC3 (LC3-II (Microtubule-associated proteins 1A/1B light chain 3A)) and degradation of p62/SQSTM1 (sequestosome 1) (Figure 2A and Appendix A). These findings are well corresponded to previously obtained data, which demonstrated the emergence of apoptosis and autophagy as a consequence of mitotic catastrophe [11,20].

The co-treatment of all tested cells with doxorubicin and pan-caspase inhibitor, zVAD-fmk, led to the switch from the apoptotic to necroptotic cell death and autophagy perturbation that was documented by the inhibition of caspase-3 processing, accumulation of lipidated LC3-II, and RIP1 phosphorylation (Figure 2A and Appendix A). To validate the induction of RIP1-dependent cell death after doxorubicin/zVAD-fmk treatment, the cells were treated with RIP1 inhibitor necrostatin-1s. The addition of necrostatin-1s prevented phosphorylation of RIP1, which proved RIP1-dependent necroptotic death in U1810, Caov4, and HCT116 cells after induction of mitotic catastrophe (Figure 2A). Moreover, immunoprecipitation with caspase-8-specific antibody at 48 h of doxorubicin/zVAD-fmk treatment revealed the interaction between phospho-RIP1, caspase-8, and FADD, which are parts of a large molecular complex for necroptosis execution also known as the Ripoptosome (Figure 2B) [21]. Thus, it provides additional confirmation that Caov4, HCT116, and U1810 cell lines treated with a 600 nM doxorubicin in combination with pan-caspase inhibitor are dying via RIP1-dependent necroptosis.

To provide additional evidence that necroptosis is a consequence of mitotic catastrophe upon doxorubicin and zVAD-fmk treatment, the phospho-form of MLKL, which is known as a biomarker and a crucial mediator of necroptosis, was analyzed by immunofluorescence. The treatment with a combination of TNF-α (Tumor necrosis factor-alpha), zVAD-fmk, and Smac mimetic BV6 was used as a positive control for necroptotic induction and MLKL phosphorylation. The untreated and doxorubicin/zVAD-fmk/necrostatin-1-treated cells were used as a negative control for phospho-MLKL immunofluorescence (Appendix A). The phosphorylation form of MLKL was detected in cells characterized by multiple nuclei/micronuclei upon doxorubicin and zVAD-fmk treatment. The treatment with TNF-α, zVAD-fmk, and BV6, a classical combination for triggering RIP-dependent necroptosis [22,23,24], also led to MLKL phosphorylation, which corroborated the specificity of the indicated marker (Figure 2C). Thus, these findings clearly revealed that necroptosis could be a novel additional consequence of mitotic catastrophe.

### 2.3. Interplay between Necroptosis, Autophagy, and Apoptosis under Conditions of Mitotic Catastrophe

Since doxorubicin-induced mitotic catastrophe could lead to apoptosis, autophagy modulation, and necroptosis, the interdependence between these different types of cell death upon mitotic catastrophe induction was investigated. The HCT116 and Caov4 cell lines were treated with 600 nM doxorubicin alone or in combination with zVAD-fmk, an inhibitor of apoptosis, and/or necrostatin-1s, an inhibitor of necroptosis, and/or autophagy inhibitor bafilomycin A1 to protect against induction of the appropriate type of cell death. As mentioned above, the doxorubicin treatment resulted in caspase-3 processing and cleavage of PARP-1, while the addition of zVAD-fmk prevented the appearance of apoptotic markers and led to accumulation of the phosphorylated form of RIP1 and necroptosis induction (Figure 3A and Appendix A). The presence of bafilomycin A1, an agent that inhibits lysosomal acidification and autophagosome-lysosome fusion, reduced degradation of the autophagy substrates LC3-II and p62/SQSTM1. Interestingly, the treatment of cells with bafilomycin A1 alone or in combination with doxorubicin significantly increased the level of caspase-3 processing and cleaved PARP-1 in HCT116 cells (Figure 3A). Treatment of Caov4 cells with bafilomycin A1 also led to the appearance of the cleaved form of PARP-1. These consequences could be completely suppressed by co-treatment with caspase inhibitor zVAD-fmk, which indicates the ability of bafilomycin A1 treatment to induce caspase-dependent apoptosis (Figure 3A). Moreover, the effect of bafilomycin-induced apoptosis was confirmed by analyzing the appearance of Sub-G1 populations. As shown in Figure 3B, the percentage of apoptotic cells with Sub-G1 DNA content was significantly increased upon treatment with bafilomycin A1 alone or in combination with doxorubicin. Importantly, the simultaneous inhibition of autophagic and apoptotic cell death by bafilomycin A1 and zVAD-fmk increased phosphorylated levels of RIP1 in both cell lines, promoting necroptosis (Figure 3C). It should be noted that autophagy is able to play a pro-survival role, however, its longtime inhibition might promote other cell death modalities, for example, necroptosis.

To minimize the effect of bafilomycin-induced apoptosis, wild type U1810 and ATG13-knockout U1810 cell lines were examined upon treatment with doxorubicin alone or in combination with zVAD-fmk and/or necrostatin-1s. ATG13-knockout cells showed a decrease of the lipidated form LC3-II in untreated cells, a strong accumulation of p62/SQSTM1 and a higher level of non-lipidated LC3-I in response to all treatments, which indicated a significant reduction in autophagic flux (Figure 3C). Compared with wild type U1810 cells, in ATG13-knockout cells, an increase in apoptotic markers upon mitotic catastrophe induction was not observed. However, the treatment with zVAD-fmk and doxorubicin resulted in a significant increase in phosphorylation of RIP1 in ATG13-knockout cells relative to the parental cell line (Figure 3C and Appendix A). The addition of selective inhibitor of necroptosis, necrostatin-1s, completely blocked RIP1 phosphorylation and suppressed necroptosis in both cell lines. Thus, our results indicate that mitotic catastrophe induction in combination with caspase inhibition could trigger the necroptotic cell death which can be accelerated by autophagy suppression.

### 2.4. Inhibition of Different Types of PCD Modulates Mitotic Catastrophe

To further examine how the modulation of different cell death programs could affect mitotic catastrophe, we evaluated cells with multinuclei/micronuclei upon doxorubicin treatment alone or in combination with zVAD-fmk, necrostatin-1s, or bafilomycin A1. The analysis of microscopic images of Caov4, HCT116, and U1810 cells (wild type or ATG13-knockout), treated with zVAD-fmk/doxorubicin, revealed a significant increase in the number of cells with mitotic catastrophe morphology upon caspase inhibition (Figure 4A). Interestingly, in Caov4 cells, the combination of RIP1 inhibitor, necrostatin-1s, with zVAD-fmk/doxorubicin treatment decreased the accumulation of cells with multinuclei/micronuclei from 36% to 22%. The suppression of autophagy with bafilomycin A1 in Caov4 and HCT116 cells reduced the mitotic catastrophe cell population upon doxorubicin treatment (Figure 4A). Importantly, concomitant treatment with zVAD-fmk and bafilomycin A1 did not increase the incidence of mitotic catastrophe and could be associated with increased necroptotic cell elimination (Figure 3A,C). In ATG13-knockout and wild type U1810 cells, we could detect only a mild decrease of the cell population characterized by mitotic catastrophe morphology under the conditions described above (Figure 4B). Taken together, all data suggest that autophagy suppression could attenuate mitotic catastrophe formation in some cell lines.

Inhibition of necroptosis led to different consequences. In Caov4, but not in HCT116 and U1810 cells, the treatment with necrostatin-1s significantly decreased the cell population with mitotic catastrophe morphology upon zVAD-fmk/doxorubicin administration (Figure 4A). These data proposed that in some tumor cell lines or tissues necroptosis suppression could modulate mitotic catastrophe formation. Altogether, these data indicate that 600 nM doxorubicin treatment combined with caspase inhibitor zVAD-fmk led to an accumulation of cells with mitotic abnormalities in which necroptotic cell death is triggered. Inhibition of autophagy and necroptosis in some cases decreases an accumulation of cells in mitotic catastrophe state that reveals the role of these PCD types in mitotic catastrophe development.

## 3. Discussion

Mitotic catastrophe is considered as a pre-stage of cell death caused by aberrant mitosis [20]. The defects in chromosomes, mitotic spindles, or the cytokinesis apparatus could promote mitotic catastrophe, which is characterized by unique nuclear morphology including the formation of micro- and/or multinuclei. According to the recommendations of the Nomenclature Committee on Cell Death, mitotic catastrophe can end in the form of apoptosis, necrosis, or senescence-mediated elimination of mitosis-deficient and genomically unstable cells preventing carcinogenesis [12]. Importantly, the type of necrotic death which is induced upon mitotic catastrophe was unknown.

In current research and in previous studies, it was demonstrated that the treatment of cells with 600 nM of the DNA-damaging drug, doxorubicin, triggers mitotic catastrophe and subsequent apoptotic death [10,11,20]. Additionally, mitotic catastrophe formation leads to autophagy modulation and autophagy-dependent cell death [10]. Notably, autophagy can help cells to avoid death or promote it in a time-dependent manner. On early step of cell stress, autophagy provides clearance from damaged organelles and supplies additional energy. On the later stage, this process promotes death, destroying important for life cell components. To identify the role of necroptosis and interplay between this and other types of PCD under mitotic abnormalities, we examined the effect of selective cell death inhibitors, such as zVAD-fmk, necrostatin-1s, and bafilomycin A1, in combination with doxorubicin treatment. We found that the co-treatment of cells with 600 nM doxorubicin and pan-caspase inhibitor, zVAD-fmk, increased the number of cells with mitotic catastrophe morphology, preventing apoptotic elimination of cells and stimulating RIP1-dependent necroptotic cell death. Simultaneously, the inhibition of effector caspase-3 processing, formation of the Ripoptosome complex, and RIP1 phosphorylation were detected (Figure 2A,B). Moreover, using confocal microscopy we demonstrated the necroptosis execution in cells with mitotic catastrophe morphology that was detected by phosphorylation of MLKL (Figure 2C). Interestingly, the recent study of Liccardi et al. showed that the Ripoptosome complex assembles during physiological mitosis, generating a pulse of sublethal caspase-8 activity [25]. Suppression of caspase activity by pan-caspase inhibitors, QVD/zVAD-fmk, reduces the availability of active PLK1 (polo-like kinase 1 also known as serine/threonine-protein kinase 13) to phosphorylate important downstream substrates required for chromosome alignment, which results in mitotic defects or appearance of multipolar spindles. These data are in agreement with our findings and explain an increase of the number of cells with mitotic catastrophe morphology upon zVAD-fmk/doxorubicin treatment (Figure 4). Moreover, the Ripoptosome formation during cell division explains necroptosis induction after mitotic catastrophe. Cells stuck in the G2/M phase accumulate this complex, which can trigger apoptosis or necroptosis depending on conditions. Furthermore, the study of Gupta and Liu confirmed the important role of the Ripoptosome formation and RIP kinases in necroptosis occurred in nocodazole-arrested cells, which represent a mitotic population, demonstrating necroptotic cell death during mitosis [26]. Thus, we could conclude that under mitotic catastrophe induction, caspase inhibition promotes an accumulation of cells with mitotic abnormalities and triggers RIP1-dependent necroptotic death of cells with mitotic catastrophe morphology. Additionally, we also observed that zVAD-mediated mitotic catastrophe was fully rescued upon co-treatment with RIP1 kinase inhibitor, necrostatin-1s, in zVAD-fmk/doxorubicin-treated Caov4 cells (Figure 4A). These observations are consistent with Liccardi’s model, which proposes that PLK1 could be recruited into the Ripoptosome complex via RIP1 and the suppression of RIP1 de-represses PLK1, leading to its activation, substrate phosphorylation, and restoration of clonogenic potential of the cells [25]. Importantly, blocking caspase activity can be achieved not only by chemical inhibition but also in response to genetic mutations. The mutations, which disturb the caspase activation or decrease their levels, could promote necroptosis induction after mitotic catastrophe in vivo without chemical inhibition. Indeed, in some human tumors, such as head and neck squamous carcinomas or hepatocellular carcinomas, initiator caspase-8 is often mutated or deleted [27,28]. Inhibition of caspase-8 or its mutations elevate the levels of chromosomal aberrations [29] and predisposes a wide variety of cancers to necroptosis [30].

The previous study from our group demonstrated that in colorectal carcinoma cells autophagy modulation is an essential step for the execution of cell death after mitotic catastrophe [11]. Therefore, here we investigated the role of autophagy in regulation of apoptotic and necroptotic cell death upon mitotic perturbations. The combined treatment with autophagic inhibitor bafilomycin A1 and doxorubicin reduced the number of mitotic catastrophe cells and enhanced apoptotic cell death. The ATG13-knockout U1810 cells demonstrated only a slight decrease of mitotic catastrophe formation in comparison with wild type cells. According to Sub-G1 analysis (Figure 3A,B), bafilomycin A1 itself stimulated apoptotic death, which would explain the decrease of the cell population in the mitotic catastrophe state. However, chemical inhibition of apoptosis in this setting did not recover the incidence of mitotic catastrophe. Consequently, in these conditions chemical or genetic blockage of the autophagic process attenuates mitotic catastrophe formation.

Interestingly, we found that in cell lines treated with bafilomycin A1 and in ATG13-knockout U1810 cells the level of RIP1 phosphorylation was significantly increased upon zVAD-fmk/doxorubicin treatment (Figure 3A,D). This observation is consistent with the majority of previous studies, which demonstrated that autophagy is able to inhibit necroptosis in various cell lines, stimulated by TNFα, antigens, or starvation [31,32]. The recent data indicate that lysosomal turnover via autophagy is critical for preventing the accumulation of active RIP kinases, and autophagy suppression could stimulate necroptotic signaling [33]. Thus, one can propose that after mitotic catastrophe, cells are prone to die via necroptosis if autophagy is downregulated (Figure 5). There is significant evidence that autophagy helps apoptosis-resistant tumor cells to survive [34]. For these tumors, inhibition of autophagy might help to improve chemotherapeutic drug outcome via stimulation of mitotic catastrophe and following necroptosis.

In summary, our findings are the first to demonstrate that mitotic catastrophe induced by a 600 nM of the DNA-damaging drug doxorubicin in combination with caspase inhibition terminates an existence of cells with mitotic abnormalities via necroptotic cell death, accompanied by the Ripoptosome complex formation and RIP1 and MLKL phosphorylation. Moreover, the chemical or genetic suppression of autophagy promotes RIP1 phosphorylation and necroptosis induction. Necroptosis induces rapid membrane permeabilization via phosphorylated MLKL protein, leading to the release of intracellular immunogenic damage-associated molecular patterns (DAMPs) that can activate innate immune pattern recognition receptors and stimulate efficient immune cell recruitment [35]. Furthermore, RIPK1-mediated induction of NF-κB and downstream target genes are necessary for initiating CD8+ T cell adaptive immunity [36]. Thus, due to the immunogenic nature of necroptosis and aneuploidy status of most tumor cells, the induction of necroptotic cell death after mitotic catastrophe could be an effective approach to overcome drug resistance to sublethal doses of DNA-damaging drug, significantly limit side effects, and stimulate the anti-tumor immune response.

## 4. Material and Methods

### 4.1. Cell Culture and Treatments

The human ovarian carcinoma Caov4 cells (ATCC, HTB-76) and human colorectal carcinoma HCT116 cells (ATCC, CCL-247) were cultured in DMEM (Dulbecco’s Modified Eagle Medium) (Gibco) supplemented with 10% (*w*/*v*) fetal bovine serum (Gibco) and 100 μg/mL penicillin, 100 μg/mL streptomycin, and 0.25 μg/mL of Fungizone (Gibco). Human lung adenocarcinoma U1810 cells (from the collection of Uppsala University, Uppsala, Sweden) and ATG13-knockout U1810 cells [37] were cultured in RPMI (Roswell Park Memorial Institute) 1640 medium (Gibco) supplemented with 10% (*w*/*v*) fetal bovine serum (Gibco), 100 μg/mL penicillin, 100 μg/mL streptomycin, and 0.25 μg/mL of Fungizone (Gibco). Cells were grown in a CO_2_ incubator (5% CO_2_) at 37 °C and maintained in a logarithmic growth phase for all experiments. Throughout the experiments, cells were stimulated with 600 nM doxorubicin (Sigma) for 24 or 48 h. To block apoptosis, necroptosis, or autophagy, cells were incubated in the presence of 40 μM caspase inhibitor zVAD-fmk (Z-Val-Ala-DL-Asp-fluoromethylketone, N-1560, Bachem, Switzerland), 30 μM RIP1 inhibitor necrostatin-1s (7-Cl-O-Nec1, ab221984, Abcam Eugene, OR, USA), and 25 nM bafilomycin A1 (B1793, Sigma-Aldrich, Saint Louis, MS, USA) for 1 h, respectively, after which 600 nM doxorubicin was added. TNF-α (Tumor necrosis factor-alpha, gift from Generium), zVAD-fmk, and Smac mimetic BV6 (N,N’-(hexane-1,6-diyl)bis(1-{(2S)-2-cyclohexyl2-[(N-methyl-L-alanyl)amino]acetyl}-L-prolyl-beta-phenyl-Lphenylalaninamide), 5.33965, Sigma-Aldrich, Saint Louis, MS, USA) were used as a positive control for necroptotic induction.

### 4.2. Gel Electrophoresis and Western Blot

Cells were harvested, washed in DPBS (Dulbecco’s Phosphate-Buffered Saline), and lysed for 10 min at 4 °C in RipA buffer (Radioimmunoprecipitation Assay buffer) supplemented with Halt protease inhibitors (78430, Thermo Scientific, Rockford, IL, USA) and Phosphatase Inhibitor Cocktail 2 (P5726, Sigma-Aldrich, Saint Louis, MS, USA). Cellular extracts were centrifuged at 13,000 rpm for 20 min at 4 °C to separate the insoluble material, followed by a determination of protein concentration using the BCA (Bicinchoninic Acid) assay (23225, Thermo Scientific, Rockford, IL, USA). Equal amounts of protein from each sample were mixed with Laemmli’s loading buffer, boiled for 5 min at 95 °C, and subjected to sodium dodecyl sulphate–polyacrylamide gel electrophoresis. Proteins were blotted onto nitrocellulose membranes with a Bio-Rad TransBlot Turbo (Bio-Rad) machine, then the membranes were transferred to a solution of 5% non-fatty milk in TBST (TBS (Tris-buffered saline) with 0.05% Tween 20) and washed in TBST three times for 5 min. The membranes were incubated with primary antibodies against cleaved PARP-1 (Asp214) (9541S), RIP1 (4926S), phospho-RIP (Ser166) (65746S), caspase-3 (9662S), cleaved caspase-3 (9664S), and ATG13 (6940S) (all from Cell Signaling Technology, Danvers, MA, USA), PARP-1 (ab137653), p62/SQSTM1 (ab56416), LC3B (ab51520), and vinculin (ab129002) (all from Abcam, Eugene, OR, USA), and FADD (gift from Prof. Inna Lavrik, Otto von Guericke University Magdeburg, Magdeburg, Germany) overnight at 4 °C. After three washes in TBST, the membranes were incubated with goat anti-rabbit (ab6721) or anti-mouse IgG-HRP antibody (ab205719)—both from Abcam—for one hour.

### 4.3. Immunoprecipitation

Cell were harvested, washed in DPBS, and lysed using Lysis buffer (Tris HCl pH 7.4 20 mM, NaCl 137 mM, EDTA 2 mM, Glycerin 10%, Triton X-100 1%) supplemented with Halt protease inhibitors (78430, Thermo Scientific, Rockford, IL, USA) and Phosphatase Inhibitor Cocktail 2 (P5726, Sigma-Aldrich, Saint Louis, MS, USA). The 1 μg of mouse monoclonal antibodies against caspase-8 and 100 μg of protein (gift from Prof. Inna Lavrik, Otto von Guericke Magdeburg University, Magdeburg, Germany or 9746S, Cell Signaling Technology, Danvers, MA, USA) were incubated overnight at 4 °C. The samples were subjected to centrifugation (2000 rpm, 5 min, 4 °C) and the supernatant was collected. The Protein G Sepharose 4 Fast Flow (GE Healthcare, Chicago, IL, USA) beads were added to cell lysates and incubated for 4 h at 4 °C with rotation. After incubation, the beads were washed two times with 1 mL of Lysis buffer and two times with DPBS. The beads were dried with a syringe and 2× Laemmli loading buffer was added to the samples for gel electrophoresis.

### 4.4. Sub-G1 Test

For FACS analysis, cells were harvested at the indicated time points, fixed in 70% ethanol overnight, and stained with 50 µg/mL propidium iodide (556463, BD Biosciences, Cayey, Puerto Rico) in the presence of 100 µg/mL RNase A (12091, Invitrogen, Merelbeke, Belgium) at 37 °C for 10 min. Flow cytometric analysis was carried out using a FACS Canto II flow cytometer equipped with BD Bioscience software (Becton Dickinson, San Jose, CA, USA).

### 4.5. Immunofluorescence Microscopy

Cells were grown, treated, fixed, and stained directly on coverslips. Next, 500 nM MitoTracker Red FM (M22425, Invitrogen, Merelbeke, Belgium) was added for 20 min. The cells were fixed for 15 min with 4% paraformaldehyde (Sigma) at room temperature, permeabilized with 0.5% Triton™ X-100 for 10 min, blocked with 2.5% BSA for 1 h, and incubated for 1 h at room temperature with primary monoclonal rabbit anti-phospho-MLKL (S358) (1:100, ab187091, Abcam, Eugene, OR, USA) antibody diluted in DPBS supplemented with 2.5% BSA with 0.05% Tween-20. Next, cells were incubated with antirabbit IgG secondary antibody Alexa Fluor^®^ 488 Conjugate (1:500, A-21206, Invitrogen, Eugene, OR, USA) for 1 h at room temperature. The counterstaining of nuclei was carried out by incubation for 10 min with 1 μg/mL DAPI (4′,6-diamidino-2-phenylindole, D1306, Invitrogen, Eugene, OR, USA) or Hoechst 33,342 (Trihydrochloride, Trihydrate, H1399, Invitrogen, Eugene, OR, USA). Between all steps, cells were washed three times for 5 min with DPBS. Stained sections were mounted using Vectashield Antifade Mounting Medium (H-1000, Vector Laboratories, Burlingame, CA, USA) and examined under an LSM 780 confocal laser scanner microscope (Zeiss). Mitotic catastrophe development was evaluated by analysis of nuclear morphology under the 63×/1.4 oil objective. At least 300 cells were counted and analyzed for mitotic catastrophe morphology for each experiment.

### 4.6. Data Processing, Statistical Analysis, and Graphic Illustrations

Western blot images were processed using Image Lab software. Densitometric analysis was performed using Image Lab Software or Image J and presented in Appendix A. For normally distributed data, the Student’s *t*-test was performed to analyze statistically significant differences between groups. *p*-values lower than 0.05 were considered statistically significant. The illustrations from Servier Medical Art [38] were reproduced under the Creative Commons License attribution 3.0 Unported License.

## Figures and Tables

**Figure 1 ijms-23-03733-f001:**
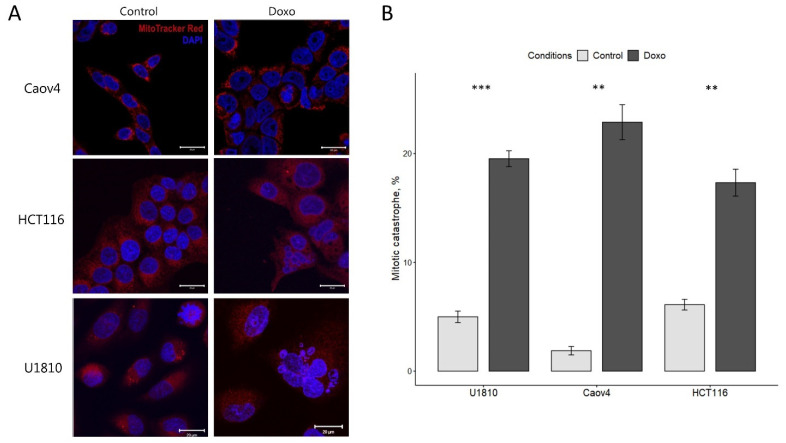
Mitotic catastrophe in doxorubicin-treated Caov4, HCT116, and U1810 cells. All cell lines were treated with 600 nM doxorubicin for 48 h and stained with MitoTracker Red FM (red) to observe mitochondria and with DAPI (blue) to detect nuclei. The cells were monitored under a confocal microscope. (**A**) Representative images of mitotic catastrophe-associated nuclear morphology. (**B**) Quantification of mitotic catastrophe after 48 h treatment with doxorubicin. The numbers of mitotic catastrophe cells examined in each cell line are shown in the bars. Values are the mean (±standard deviation of the mean) of three independent experiments. Control: no treatment; Doxo: Doxorubicin. ** *p* < 0.01, *** *p* < 0.001.

**Figure 2 ijms-23-03733-f002:**
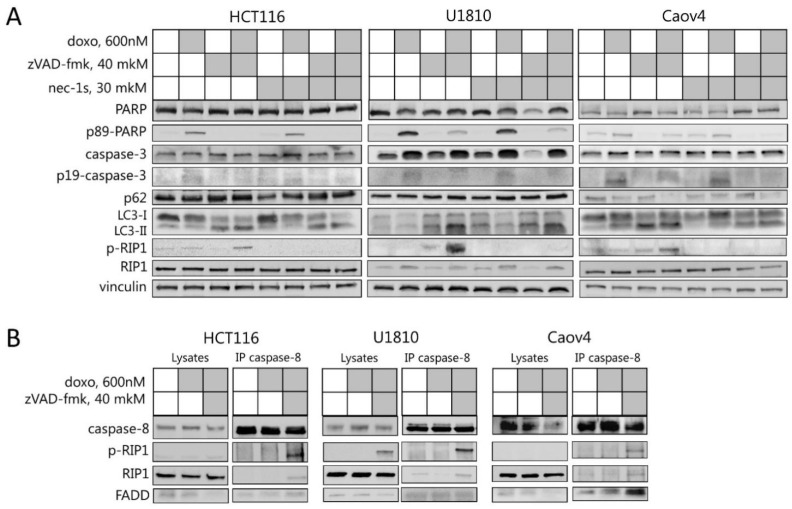
Necroptosis execution in cells upon mitotic catastrophe induction. (**A**) Caov4, HCT116, and U1810 cells were pre-treated with 40 μM zVAD-fmk and/or 30 μM necrostatin-1s then treated for 48 h with 600 nM doxorubicin. Immunoblot analysis using the indicated antibodies is shown. (**B**) Caov4, HCT116, and U1810 cells were treated with 600 nM doxorubicin and 40 μM zVAD-fmk. Lysates from untreated and treated cells were immunoprecipitated with anti-caspase-8 antibodies. Immunoblot analysis using the indicated antibodies is shown. (**C**) Immunofluorescence analysis using anti-phospho-MLKL antibodies in Caov4, HCT116, and U1810 cells treated with the indicated agents. Green dots indicate phospho-MLKL. Cell nuclei were counterstained with Hoechst 33342 (blue). Scale bars: 10 μm.

**Figure 3 ijms-23-03733-f003:**
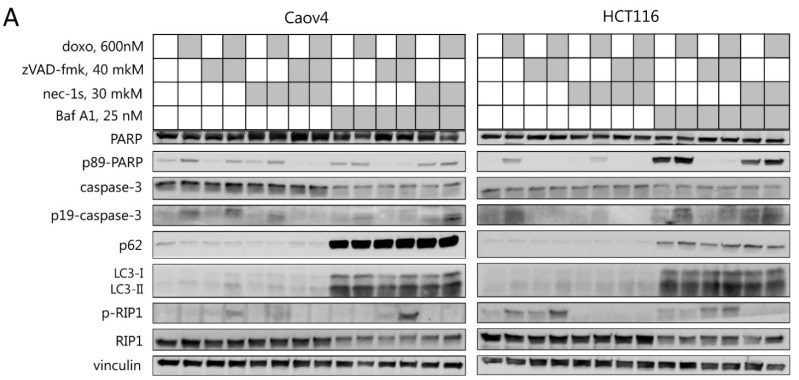
Crosstalk between necroptosis, autophagy, and apoptosis after mitotic catastrophe induction. (**A**) Caov4 and HCT116 cells were treated with 600 nM doxorubicin and 40 μM zVAD-fmk, and/or 30 μM necrostatin-1s, and/or 25 nM bafilomycin A1 for 48 h. Immunoblot analysis using the indicated antibodies is shown. (**B**) Sub-G1 analysis of Caov4 and HCT116 cells treated with 600 nM doxorubicin and 40 μM zVAD-fmk, and/or 30 μM necrostatin-1s, and/or 25 nM bafilomycin A1 for 48 h. Values are the mean (±standard deviation of the mean) of three independent experiments. (**C**) U1810 wild type and ATG13-knockout cells were treated with 600 nM doxorubicin and 40 μM zVAD-fmk and/or 30 μM necrostatin-1s for 24 h. Immunoblot analysis using the indicated antibodies is shown. * *p* < 0.05, ** *p* < 0.01.

**Figure 4 ijms-23-03733-f004:**
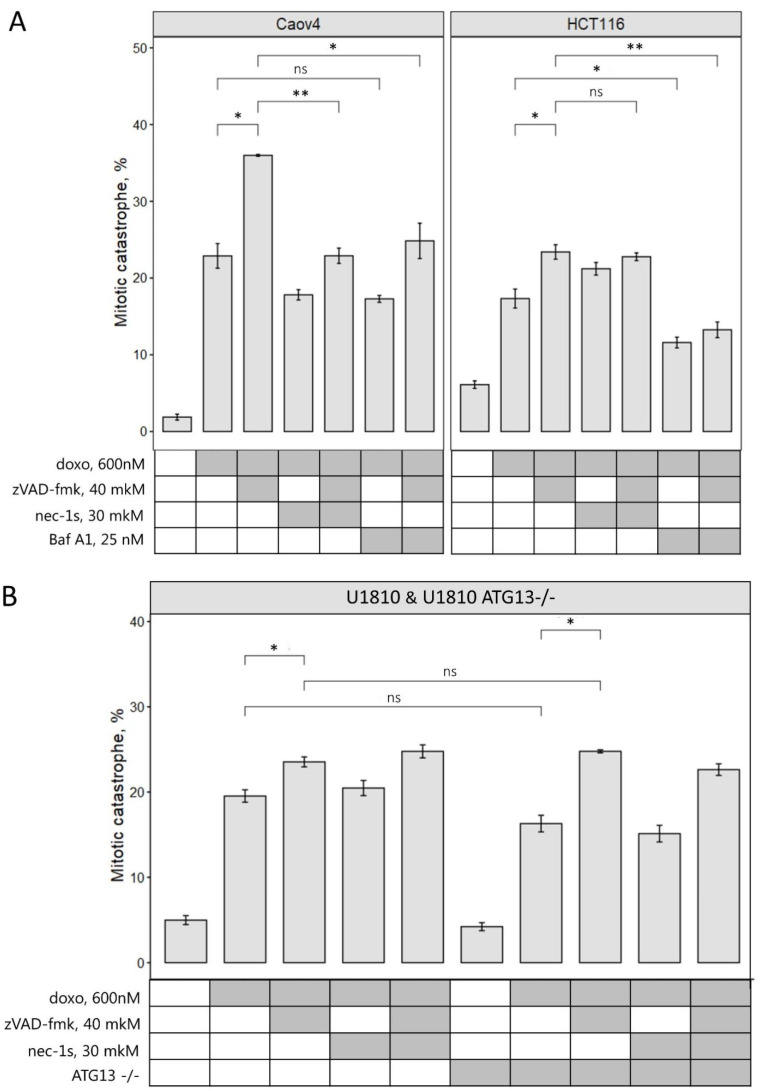
Caspase inhibition increases the population of cells with mitotic catastrophe morphology. (**A**) Quantification of mitotic catastrophe after 48 h treatment with 600 nM doxorubicin in combination with 40 μM zVAD-fmk, and/or 30 μM necrostatin-1s, and/or 25 nM bafilomycin A1 in Caov4 and HCT116 cells. (**B**) Quantification of mitotic catastrophe after 24 h of treatment with 600 nM doxorubicin in combination with 40 μM zVAD-fmk, and/or 30 μM necrostatin-1s in wild type and ATG13-knockout U1810 cells. The number of mitotic catastrophe cells examined in each cell line are shown in the bars. Values are the mean (±standard deviation of the mean) of three independent experiments. * *p* < 0.05, ** *p* < 0.01, ns—not significant.

**Figure 5 ijms-23-03733-f005:**
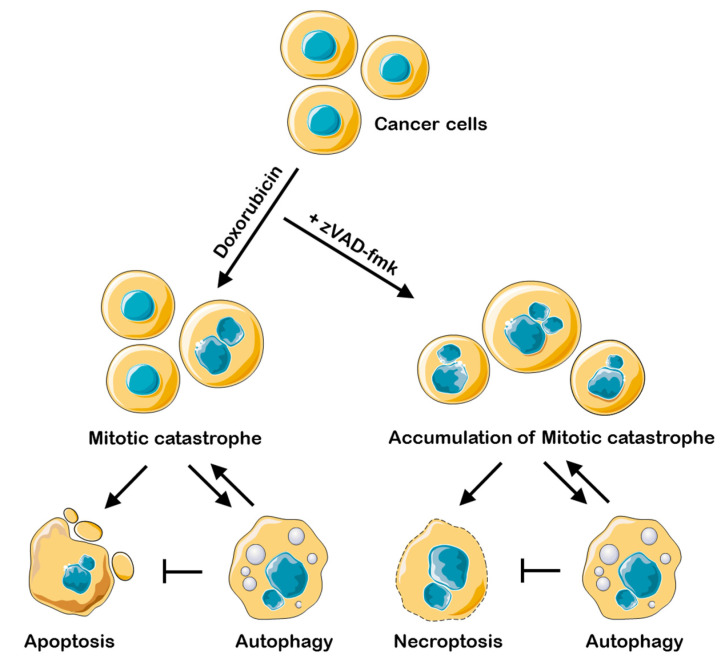
Interplay between necroptosis, autophagy, and apoptosis under conditions of mitotic catastrophe upon doxorubicin/doxorubicin-zVAD-fmk treatment. Doxorubicin treatment leads to mitotic catastrophe formation, which can be terminated by apoptosis. If catalytic activity of caspases is blocked, cells accumulate in the state of mitotic catastrophe and die via necroptosis. The development of mitotic catastrophe triggers autophagy modulation inhibition of which promotes apoptosis and necroptosis.

## Data Availability

All data generated are included in the manuscript.

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
