# Peer review of "Necroptosis as a Novel Facet of Mitotic Catastrophe"

_ijms, 2022, doi:10.3390/ijms23073733_

Round 1
Reviewer 1 Report
The manuscript by Aleksandra Yu. Egorshina and colleagues aimed at study necroptosis as a novel consequence of mitotic catastrophe induced by low-dose treatment with doxorubicin in different cancer cell lines. The manuscript is nice and well written. I think it provides valuable mechanistic insights into mitotic catastrophe induction in cancer.
However, there are a few points that need to be addressed and/or properly discussed in the paper to improve its soundness:
- In the Introduction, the authors should present the ripoptosome complex and explain its involvement in cancer cell death; moreover, since mitotic catastrophe is a mechanism promoted also by other DNA-damaging agents, including drugs candidates for repurposing in cancer therapy (for example, see on this topic the following articles: 10.1111/jphp.12503, 10.1038/s41598-021-02503-9, 10.3390/cancers11122042), the authors may expand the background on the topic by adding additional reports;
- Which are doxorubicin dosages commonly employed in clinical settings? And which are doxorubicin concentrations commonly used in in vitro experiments?
- The authors should provide evidence that low concentrations of doxo are non-toxic for normal cells, by comparing the effects of the drug on cancer cells with those observed in normal cell lines;
- Please, provide Figure 1 at high resolution, as shown in supplementary materials;
- The authors refer to the potential immune-stimulating effect of necroptosis to overcome drug resistance; it’s a very fascinating speculation that deserves at least further discussion;
- Figure 5 is graphically nice, but its meaning is not so clear and figure legend does not help.
Author Response
Response:
Dear Sir/M-me,
Thank you for high evaluation of our MS and for all your valuable comments. Below, please, find, the answers to your specific comments.
Comment 1.
In the Introduction, the authors should present the ripoptosome complex and explain its involvement in cancer cell death; moreover, since mitotic catastrophe is a mechanism promoted also by other DNA-damaging agents, including drugs candidates for repurposing in cancer therapy (for example, see on this topic the following articles: 10.1111/jphp.12503, 10.1038/s41598-021-02503-9, 10.3390/cancers11122042), the authors may expand the background on the topic by adding additional reports.
Response:
Thank you for this suggestion. Indeed, the treatment with different microtubule poisons can also lead to mitotic catastrophe. Upon the stress of microtubule network and the cell cycle arrest the Ripoptosome complex might be formed in the cell and trigger cell death. In the revised version we have extended the part related to Ripoptosome complex formation and necroptosis as well as their role in cancer cell death and inflammation. Moreover, we included suggested reference which describes the other DNA-damaging agents, including drugs candidates for cancer therapy.
Comment 2:
Which are doxorubicin dosages commonly employed in clinical settings? And which are doxorubicin concentrations commonly used in in vitro experiments?
Response:
According to different publications the plasma concentration of Doxorubicin is fluctuating between 600 and 10 ng/ml (1mkM and 18 nM, correspondently) depending on time after drug administration (PMID: 32921983, PMID: 20688160). Notably, in in vitro studies doxorubicin was used in a broad range from 0.1 to 10 mkM (for example, PMID: 30202239, PMID: 31331001). In our study we have used 600 nM of this drug.
Comment 3:
The authors should provide evidence that low concentrations of doxo are non-toxic for normal cells, by comparing the effects of the drug on cancer cells with those observed in normal cell lines;
Response:
We have tested doxorubicin on the human kidney cell line HEK293T and did not detect an essential processing of effector caspase-3 and cleavage of its substrate PARP (the most known apoptotic marker) indicating that the doxorubicin treatment at this concentration did not induce significant death in the non-tumor cell line (Fig S1).
Comment 4:
Please, provide Figure 1 at high resolution, as shown in supplementary materials;
Response:
Fig 1 has been corrected.
Comment 5:
The authors refer to the potential immune-stimulating effect of necroptosis to overcome drug resistance; it’s a very fascinating speculation that deserves at least further discussion.
Response:
Thank you for this valuable suggestion. We have included an additional discussion on this topic.
Comment 6:
Figure 5 is graphically nice, but its meaning is not so clear and figure legend does not help.
Response:
Based on your suggestion we have corrected Figure 5 and rewritten the legend.

Reviewer 2 Report
The manuscript by Aleksandra Yu. et.al concerns cancer cell fate after induction of mitotic catastrophe (MC) by DNA –damaging agent, doxorubicin. The authors used three different cell lines, to show that MC can lead to apoptotic or autophagy-like cell death but when these processes are inhibited, MC directs cells to necroptosis. Moreover, they documented that inhibition of autophagy promotes necroptosis. This is very interesting paper, but I must admit that making conclusions on the basis of representative WBs can lead to over interpretation of the results. It would be much pertinent to show densitometry analysis with the proper statistic. Moreover, it would be much easier to follow differences in the protein levels looking on graphs (representative WBs could be presented e.g. as supplementary data). Also some visualization of cells dying by different mode of cell death will make the ms more attractive.
More comments:
- In should be mentioned that, in contrast to cell death, senescence is prosurvival and can be the consequence of MC. Recently, Salmina et al. (2020) showed that upon doxorubicin treatment breast cancer cells underwent mitotic slippage, senescence, polyploidization/depolyploidization. Although the authors of this paper didn’t mention about MS, the cell morphology resembles those presented in the ms. There are also many other papers describing the phenomenon of cancer cell polyploidization/senescence upon DNA damage (see e.g special issue of Seminars in Cancer Biology: DOI: 10.1016/j.semcancer.2021.10.006). Moreover, autophagy can be not only lethal, but also cytoprotective.
- The authors mentioned the connection between Bcl -2 protein family and autophagy. What exactly did they mean?
- Autophagy per se is not a PCD. It is better to use the term “autophagy-dependent cell death”.
- There are some papers demonstrating that doxorubicin causes autophagy inhibition, even in the lower dose then used by the authors. (Nota bene I advise not to use “low dose of doxorubicin” as it is not a low dose). Doxorubicin as a low basic compound enters lysosomes and affects their acidity and functionality that in consequence leads to autophagy flux inhibition. In this context additional autophagy inhibition by Bafilomycin A1 may enhance cell death what was demonstrated by the authors. However no autophagy inhibition by doxorubicin was demonstrated. How it could be explained?
- I cannot see any decrease in lipidated form of LC3 in doxorubicin- treated U1810 cells, both wide type and ATG13 KO cells, except the case when cells were treated with doxorubicin, zVAD-fmk and nec-1s together. Decrease in LC3II ( lipidated) form is visible only in untreated cells. Moreover decrease in the level of LC3 II form is not connected with autophagy flux inhibition, but in contrary, its elevation accompanied with p62 increase is associated with autophagy flux inhibition.
- In Fig 1A in doxorubicin-treated CaoV4 the slight elevation of LC3 II and decrease of p62 protein levels are visible. Even if changes in LC3II and p62 may suggest autophagy activation, there is no prove that this activation leads to autophagy dependent-cell death. If cell death is a result of autophagy activation, its inhibition (e.g. by Bafilomycin ) should prevent it. However, in Fig 3 A-C is shown that Bafilomycin did not prevent induced cell death because there is no significant changes in cleaved form of PARP and caspase 3. Moreover, Bafilomycin significantly increased SubG1 in doxorubicin-treated Caov4 and HCT cell line. These results strongly suggest that autophagy activation evoked by doxorubicin can have pro-survival effect. By the way, as the authors show subG1, they probably have cell cycle analysis by flow cytometry. It would be nice to see also G2/M phase.
- Line 272-273. If Ripoptosome affect mitotic catastrophe why no changes in cell number with mitotic catastrophe after necrostatin-1s that is an inhibitor of RIP1 are visible? ( Fig 4A , B). A slight decrease is only shown in Caov4 cells. Necrostatin-1s prevented only zVAD-fmk-caused elevation of the cell number with mitotic catastrophe morphology. However this result was restricted only to Caov4 cell line.
- Line 298-300. Chemically or genetically inhibited autophagy reduced MC only in case of HCT 116 but not Caov4 and U1810 cell lines ( Fig 4Aand B). Thus, it should be not generalized.
- In figure legends the info how many independent experiments were done should be included.
Author Response
Dear Sir/M-me,
Thank you for the important questions and suggestions. We have done densitometric analysis and included these results in the Supplementary part (Fig S2). As to visualization of the different death modes, unfortunately, there are some problems. For example, we cannot use microscopy due to detachment of dead cells from cover glasses. Flow cytometry is not able to distinguish late apoptotic and necrotic/necroptotic populations, because Annexin V-FITC and propidium iodide (the most popular dyes for detection of cell death) penetrate the plasma membrane during both necrosis and late stage of apoptosis. We tried to visualize the different modes of cell death using imaging flow cytometry to estimate apoptotic and necroptotic populations as we had shown previously (PMID: 30025878). Unfortunately, again we could not distinguish necrotic and late apoptotic cell populations because doxorubicin overlayed with propidium iodide and the nucleus morphology was significantly changed upon mitotic catastrophe. Below, please, find, the answers to your specific comments.
Comment 1:
In should be mentioned that, in contrast to cell death, senescence is prosurvival and can be the consequence of MC. Recently, Salmina et al. (2020) showed that upon doxorubicin treatment breast cancer cells underwent mitotic slippage, senescence, polyploidization/depolyploidization. Although the authors of this paper didn’t mention about MS, the cell morphology resembles those presented in the ms. There are also many other papers describing the phenomenon of cancer cell polyploidization/senescence upon DNA damage (see e.g special issue of Seminars in Cancer Biology: DOI: 10.1016/j.semcancer.2021.10.006). Moreover, autophagy can be not only lethal, but also cytoprotective.
Response:
We agree that DNA-damaging agents can trigger a broad spectrum of cell responses and not only mitotic catastrophe or cell death. In the revised version we have added discussion concerning a link between DNA damage, mitotic catastrophe and senescence. In many cases autophagy might help cells to escape death; however, previously several groups including our demonstrated that mitotic catastrophe might stimulate cell death via autophagy (PMID: 29109414).
Comment 2:
The authors mentioned the connection between Bcl -2 protein family and autophagy. What exactly did they mean?
Response:
We have rewritten this part and explained this link.
Comment 3:
Autophagy per se is not a PCD. It is better to use the term “autophagy-dependent cell death”.
Response:
We agree with this statement. Autophagy can help cells to avoid death or promote it in a time-dependent manner. On earlier steps of cell stress, autophagy provides clearance from damaged organelles and supplies additional energy. On the later stage, this process could promote death, destroying cell components important for life. Previously using life-time microscopy we demonstrated that after development of mitotic catastrophe cells underwent autophagy-dependent cell death (PMID: 29109414). Consequently, to avoid unproved interpretation in the revised version we used the term “autophagy” and noted its dual role.
Comment 4:
There are some papers demonstrating that doxorubicin causes autophagy inhibition, even in the lower dose then used by the authors. (Nota bene I advise not to use “low dose of doxorubicin” as it is not a low dose). Doxorubicin as a low basic compound enters lysosomes and affects their acidity and functionality that in consequence leads to autophagy flux inhibition. In this context additional autophagy inhibition by Bafilomycin A1 may enhance cell death what was demonstrated by the authors. However no autophagy inhibition by doxorubicin was demonstrated. How it could be explained?
Response:
According to different studies, doxorubicin treatment causes a modulation of autophagy that induces various physiological outcome, for example, cardiac toxicity (PMID: 34081265). Notably, doxorubicin can stimulate or inhibit autophagy depending on tissue specificity and time of incubation (PMID: 19901028, PMID: 30765730). Different molecular pathways can be involved in these situations. Thus, doxorubicin treatment can lead to upregulation of Beclin-1 and LC3 level stimulating autophagy (PMID: 31452719) but affect AMPK/mTOR pathway that promote opposite effect (PMID: 32034646). According to Western blot and densitometric analysis, the treatment of all three cell types with doxorubicin led to LC3II and p62 perturbation indicating autophagy modulation (Fig 2 and Fig S2) which is consistent with the published data. Additionally, we have changed "low Doxorubicin concentration" to “600 nM of doxorubicin”.
Comment 5:
I cannot see any decrease in lipidated form of LC3 in doxorubicin- treated U1810 cells, both wide type and ATG13 KO cells, except the case when cells were treated with doxorubicin, zVAD-fmk and nec-1s together. Decrease in LC3II (lipidated) form is visible only in untreated cells. Moreover decrease in the level of LC3 II form is not connected with autophagy flux inhibition, but in contrary, its elevation accompanied with p62 increase is associated with autophagy flux inhibition.
Response:
We agree that decrease of LC3II in ATG13 KO vs wild-type cells was detected in untreated cells. PMID: 29109414However, the levels of p62 and LC3I (non-lapidated form) were significantly increased upon all treatments (Fig 3C, Fig S2, present study). This fact indicates that LC3I was not converted into LC3II. This was accompanied by the accumulation of p62, confirming autophagy flux inhibition.
Comment 6:
In Fig 1A in doxorubicin-treated CaoV4 the slight elevation of LC3 II and decrease of p62 protein levels are visible. Even if changes in LC3II and p62 may suggest autophagy activation, there is no prove that this activation leads to autophagy dependent-cell death. If cell death is a result of autophagy activation, its inhibition (e.g. by Bafilomycin ) should prevent it. However, in Fig 3 A-C is shown that Bafilomycin did not prevent induced cell death because there is no significant changes in cleaved form of PARP and caspase 3. Moreover, Bafilomycin significantly increased SubG1 in doxorubicin-treated Caov4 and HCT cell line. These results strongly suggest that autophagy activation evoked by doxorubicin can have pro-survival effect. By the way, as the authors show subG1, they probably have cell cycle analysis by flow cytometry. It would be nice to see also G2/M phase.
Response:
We agree that autophagy could play a pro-survival role upon genotoxic stress. However, previously using life-time fluorescent microscopy we demonstrated that doxorubicin-induced mitotic catastrophe led to autophagy-dependent cell death (PMID: 29109414). Moreover, if autophagy only protects cells from death in the state of mitotic catastrophe its inhibition would lead to decrease of this cell population. Nevertheless, we detected mitotic catastrophe reduction only in HCT116 cells upon autophagy blockage and not in Caov-4 or U1810 cells (Fig 4). Since Bafilomycin possesses toxic effect itself, we used the cell line with genetic inhibited autophagy (KO ATG13 U1810). These cells did not demonstrate a decrease in mitotic catastrophe population in comparison with the wild-type cells. Thus, in general autophagy does not demonstrate essential pro-survival effect for cells in the state of mitotic catastrophe. The interplay between autophagy, apoptosis and necroptosis is a multimodal process. Yet, our study demonstrates that mitotic catastrophe can be terminated by RIP1-dependent necroptosis. In this conditions autophagy inhibition leads to necroptosis acceleration which describes a new aspect of the above-mentioned multimodal interplay.
Comment 7:
Line 272-273. If Ripoptosome affect mitotic catastrophe why no changes in cell number with mitotic catastrophe after necrostatin-1s that is an inhibitor of RIP1 are visible? ( Fig 4A , B). A slight decrease is only shown in Caov4 cells. Necrostatin-1s prevented only zVAD-fmk-caused elevation of the cell number with mitotic catastrophe morphology. However this result was restricted only to Caov4 cell line.
Response:
This difference can result from the cell line specific effect. In our previous studies we have detected that Caov-4 cell line tends to necroptosis more than other cell lines (unpublished data). So, an inhibition of this pathway can have more impact on mitotic catastrophe. We highlighted this fact in the manuscript.
Comment 8:
Line 298-300. Chemically or genetically inhibited autophagy reduced MC only in case of HCT 116 but not Caov4 and U1810 cell lines ( Fig 4Aand B). Thus, it should be not generalized.
Response:
Thank you for this comment. We agree that inhibition of autophagy statistically (p<0.05) reduced mitotic catastrophe only in HCT-116 cells upon doxorubicin treatment; however, in U1810 and Caov4 cells we also detect a decrease in mitotic catastrophe population but with p-value <0.1. Moreover, the combined treatment of Caov-4 cells with DOX/zVAD-fmk/Bafilomycin led to a significant drop of mitotic catastrophe in comparison to DOX/zVAD-fmk. This fact suggests that Caov-4 cells in this condition died via necroptosis faster than other cell lines. Consequently, the interplay between mitotic catastrophe and different cell death modalities (apoptosis, autophagy or necroptosis) is very multiplex and depends on the cell line or tissue specificity.
Comment 9:
In figure legends the info how many independent experiments were done should be included.
Response:
We have corrected the figure legends.

Round 2
Reviewer 1 Report
The authors answered all my questions and the manuscript substantially improved its soundness, thus in my opinion it is now acceptable in the present form
Author Response
Thank you for high evaluation of the revised manuscript.